# Sex Differences in the Associations of Nutrient Patterns with Total and Regional Adiposity: A Study of Middle-Aged Black South African Men and Women

**DOI:** 10.3390/nu13124558

**Published:** 2021-12-20

**Authors:** Tshifhiwa Ratshikombo, Julia H. Goedecke, Melikhaya Soboyisi, Clement Kufe, Caroline B. T. Makura-Kankwende, Maphoko Masemola, Lisa K. Micklesfield, Tinashe Chikowore

**Affiliations:** 1South African Medical Research Council/WITS Developmental Pathways for Health Research Unit (DPHRU), Department of Paediatrics, School of Clinical Medicine, Faculty of Health Sciences, University of the Witwatersrand, Johannesburg 2000, South Africa; chichifhiwa@gmail.com (T.R.); julia.goedecke@mrc.ac.za (J.H.G.); melikhaya.soboyisi@gmail.com (M.S.); clekufe@gmail.com (C.K.); carolmakura@gmail.com (C.B.T.M.-K.); maphokomasemola@gmail.com (M.M.); Lisa.Micklesfield@wits.ac.za (L.K.M.); 2Non-Communicable Diseases Research Unit, South African Medical Research Council, Cape Town 7505, South Africa

**Keywords:** nutrient patterns, obesity, sex differences, total and regional adiposity, South Africa

## Abstract

The study evaluated the association between nutrient patterns with body fat and regional adiposity in middle-aged black South African (SA) men and women and determined if this differed by sex. Body fat and regional adiposity (dual-energy x-ray absorptiometry), and dietary intake (7-day quantified food frequency questionnaire) were measured in black SA men (*n* = 414) and women (*n* = 346). Using principal component analysis, nutrient patterns were computed from 25 nutrients in the combined sample. Four nutrient patterns were extracted, explaining 67% of the variance in nutrient intake. Animal and fat, as well as the vitamin C, sugar, and potassium driven patterns, were positively associated with total adiposity. In contrast, the retinol and vitamin B12 pattern was associated with the centralisation of fat. Notably, the strength of the association between the animal-driven nutrient pattern and BMI was greater in men (1.14 kg/m^2^, 95% CI (0.63–1.66)) than in women (0.81 kg/m^2^, 95% CI (0.25–1.36)) (P*_int_* = 0.017). In contrast, the plant-driven pattern was associated with higher abdominal subcutaneous adipose tissue (SAT) in women (44 cm^2^, 95% CI (22–67)) but not men (P*_int_* = 1.54 × 10^−4^). These differences suggest that although men and women have similar nutrient patterns, their associations with the whole body and regional body fat are different.

## 1. Introduction

Obesity is a major global health challenge, increasingly affecting developing countries [1]. South Africa has the highest prevalence of obesity in Sub-Saharan Africa [2,3,4], with more women than men (68% vs. 31%) presenting with overweight or obesity [5]. Additionally, middle-aged South African men and women (45–65 years) have the highest obesity rates compared to other age groups [5]. High body mass index (BMI) is associated with noncommunicable diseases (NCDs), such as cardiovascular disease (CVD), hypertension, atherosclerosis [6], type 2 diabetes, some cancers and premature death [7,8].

Obesity is typically characterised by BMI [9]. However, BMI does not measure body fat distribution, a key risk factor for non-communicable diseases (NCDs) [10,11]. Central (abdominal) fat, in particular, visceral adipose tissue (VAT), is thought to be the most critical determinant of cardiometabolic risk, whereas peripheral (gluteo-femoral) fat mass is associated with reduced cardiometabolic risk [12]. Dual-energy X-ray absorptiometry (DXA) provides an objective measure of total and regional adiposity, including estimates of abdominal VAT and subcutaneous adipose tissue (SAT), with greater precision, accuracy and objectivity than anthropometric measures [13,14,15]. Notably, women accumulate a higher percentage of body fat, greater peripheral fat mass, less central fat mass and VAT than men [16]. However, with increasing age, adiposity and menopause, fat redistribution occurs in women resulting in the accumulation of more VAT [17]. It is essential to understand the factors associated with sexual dimorphism of obesity and body fat distribution in middle-aged adults, particularly in Africa, where few objectively measured body fat and its distribution have been conducted [18].

Factors associated with obesity include physical inactivity, sedentary behaviours, socioeconomic status, diet [19,20,21], and urbanization [22]. With urbanisation, changes in transportation and work activities have reduced physical activity with a consequent reduction in energy expenditure [23,24]. Further, globalisation and urbanisation have increased access to ultra-processed and energy-dense fast foods [23,25]. These foods are high in fat and sugar and are associated with obesity [23,25]. Indeed, a study in South Africa reported that close to half (48.5%) of South Africans that consumed fast foods were obese [26].

The nutrient pattern analysis for assessing the role of the overall diet emerged as an alternative to traditional approaches that focus on single nutrients [27]. This is advantageous as people consume foods and nutrients in the context of an overall diet. Therefore, it is pivotal to evaluate the impact of the whole diet on health outcomes. Due to differences in cultures and foods consumed by people of varying ethnicities, the nutrient pattern analysis approach is now favoured as applicable and comparable across different population groups [27]. Within South Africa, some studies have shown associations between nutrient patterns and obesity in adolescents [28] and women alone [29]. However, no studies have evaluated sex differences in nutrient patterns and determined whether the associations between nutrient patterns and total and regional adiposity differ between men and women. Therefore, this study aimed to evaluate the association between nutrient patterns and DXA-derived body fat and regional adiposity in middle-aged black South African (SA) men and women and determine if this differed by sex.

## 2. Materials and Methods

### 2.1. Study Population

This cross-sectional study includes the follow-up data of the longitudinal AWI-Gen (Africa Wits-IN-DEPTH partnerships for Genomic Research) study. Phase 1 data collection was carried out from 2011 and 2015 on black South African men (*n* = 1027) and women (*n* = 1008) residing in Soweto, South Africa [30]. Follow-up data on a sub-sample of 502 men and 527 women randomly selected from the original sample was collected from January 2017 to August 2018. Pregnant and lactating women were excluded from this study. We excluded men and women without DXA data (*n* = 44), weight or height measures (*n* = 34) and physical activity data (*n* = 240) from the final analysis. Consequently, 760 (414 men and 346 women) participants had complete data after quality control and were included in the data analysis.

### 2.2. Demographic, Socio-Economic and Health Information

Demographic characteristics, socioeconomic status and health information of the participants was collected via interviewer-administered questionnaires. This information included age, sex, marital status (categorised as either single, married, divorced, living with a partner or widowed), education (classified as no formal or primary, secondary or tertiary education), employment status (categorised as employed or non-employed), smoking status (categorised as smoker or non-smoker) and HIV status (categorised as HIV positive or HIV negative). Participants not HIV positive completed an HIV antibody test (Wondfo One Step HIV–1/2 Whole Blood/Serum/Plasma: Test 2 lines Guanghu Wondfo Biotech Co., Ltd., Hong Kong, China). If participants tested positive, they were referred to a local HIV clinic but retained in the study.

### 2.3. Body Composition and Body Fat Distribution Measurements

Height and weight were measured wearing light clothing and without shoes, from which BMI (weight (kg)/height (m)^2^) was calculated. Weight was measured to the nearest 0.1 kg using an electronic weight scale (Tanita (TBF-410GS Total Body Composition Analyzer) Tokyo, Japan,) and height was measured to the nearest 0.1 cm using a wall-mounted stadiometer (Holtain, Ltd., Crosswell, UK). DXA measured the sub-total (total minus head) fat mass and fat-free soft tissue mass (Hologic QDR 4500A, Hologic Inc., Bedford, MA, USA) and analysed using APEX software version 4.0.2 (Hologic Inc., Bedford, MA, USA). Regional adiposity was characterised as android and gynoid fat mass and expressed as a percentage of sub-total fat mass (% fat mass (FM)), as previously described [31]. Abdominal VAT and SAT areas were estimated using DXA, which has been shown to perform a clinical read of VAT from a computed tomography scan [15].

### 2.4. Physical Activity, Sedentary Time and Energy Expenditure

The activPAL (PAL Technologies, Glasgow, Scotland) accelerometer which has been validated to assess physical activity, was used determined daily steps and sitting time [32]. The participants wore the ActivPAL, fitted on their left thigh to detect limb position using an inclinometer, on average for five to seven days (5.93 ± 0.86). The data was sampled at 10 Hz in 15-s epochs. The data was downloaded from the device and processed using ActivPAL 3M software. In addition, metabolic equivalents (METS) per day were extracted from the ActivPAL, which was then converted to kcal by multiplying by body weight (1 MET = 1 kcal/kg body weight) to determine the energy expenditure and used to assess the dietary energy intake reporting status described below.

### 2.5. Dietary Intake

Dietary intake was estimated using a seven-day quantitative food frequency questionnaire (QFFQ) consisting of 214 commonly eaten foods derived from analyses of eleven dietary surveys conducted in rural and urban South Africa since 1983 [33,34]. To facilitate recall during the QFFQ, flashcards with photographs of the food items were utilised to capture the previous seven-day self-reported dietary intake. Participants separated the food flashcard piles; the first pile included food items rarely/never consumed; the second pile included food items consumed less frequently (occasional); the last (third) pile had food items consumed in the past seven days. The interview then focused on items consumed during the past seven days for the QFFQ.

Portion sizes were estimated using household measures and a combination of two-dimensional life-size drawings of foods and utensils and three-dimensional food models, as described and validated by Steyn and others [35]. Coding involved converting the household measures (e.g., one cup/one serving spoon/one slice) to grams. The quantity and frequency of the regular food items consumed were recorded and expressed in g/day to allow for the average intake to be calculated over the previous seven days. Nutrient composition (energy and macronutrients) was calculated from the conversion of single food items to nutrient compositions, using the nutrient analysis software FoodFinder3 (SAMRC, Cape Town, South Africa), based on South African food composition tables hosted by South African Medical Research Council (SAMRC) [36].

Energy intake reporting status was evaluated as the ratio of energy intake: energy expenditure. The energy intake was derived from the QFFQ, and energy expenditure was computed from accelerometry (activPAL). Energy intake: energy expenditure ratio was used to categorise plausible reporters (0.7–1.42), under-reporters (<0.7), and over-reporters (>1.42) [31]. Energy intake reporting status was used in all subsequent analyses.

### 2.6. Statistical Analysis

Data analysis was completed using the statistical package for social scientists (SPSS) version 23 (IBM, Chicago, IL, USA). Normality tests for the continuous variables were performed using quantile-quantile plots (Q-Q plots) and the Shapiro–Wilk test. The descriptive data of the participants were stratified by sex and compared using Chi-square tests for categorical data, *t*-test for continuous parametric data and the Kruskal–Wallis test for non-parametric continuous data.

Principal component analysis (PCA) was used to extract the nutrient patterns based on the QFFQ-derived intake of 25 nutrients. Of these 25 nutrients, total protein was split into animal protein and plant protein; total carbohydrates into total sugar, starch, and total dietary fibre; and total fat categorised as saturated, monounsaturated and polyunsaturated fat. Alcohol was not considered as a nutrient and not included in deriving the nutrient pattern analysis. The nutrient intake variables were log-transformed to remove bias due to variance of the different measures of scale used to quantify the nutrients. Log transformation gives an advantage as it renders the variances and covariances independent of scale. To control for the variability of nutrient intakes from variation in energy intake, nutrients (log variables) were adjusted for log total energy intake before applying the PCA using the nutrient density approach suggested by Willet [32]. PCA was performed with the variance based on the covariance matrix and Varimax rotation. The PCA was a suitable data reduction approach for the nutrient data in this study, as was indicated by a Kaiser–Meyer–Olkin measure of sampling adequacy of 0.911 and a significant (*p* < 0.001) Bartlett’s test of sphericity. PCAs were first performed separately for both sexes. However, as the men and women had similar nutrient patterns, the two groups were combined for all subsequent analyses. The retained patterns were determined considering several criteria, including the interpretation of the patterns, the percentage of total variance explained and visual inflexions in the scree-plot (Figure 1) [35]. Nutrient patterns were named using the nutrients with absolute loadings greater or equal to 0.47 [28,37]. Nutrients with positive loadings were significantly associated with a nutrient pattern, while negative loadings were negatively associated.

Multivariable linear regression models were used to explore the associations between the nutrient pattern and measures of total and regional adiposity, namely BMI, total fat mass (%), android and gynoid fat mass (% FM), VAT (cm^2^) and SAT (cm^2^) in the combined sample of men and women. In each multivariable model, the four nutrient patterns were included as independent predictors in the same model and the covariates included sex, age, number of steps, sitting time, dietary energy intake reporting status (under, plausible and over-reporters), educational achievement, marital status, smoking, and HIV status. For VAT, body fat (kg) was also used as a covariate. To determine whether the relationships between nutrient patterns with total and regional adiposity differed by sex, we repeated the regression analyses, introduced a nutrient pattern x sex interaction term into the models, and presented any significant interactions as sex-stratified analysis. The statistical significance was set at *p* < 0.008 (0.05/6 (number of multivariate linear regression tests)) to correct for bias due to multiple testing using the Bonferroni correction.

### 2.7. Power Calculation

Sample size estimation of factor analysis is a highly debated topic [38]. However, we estimated the sample size using the guidelines recommended Mundfrom et al., 2005 and Guilford, 1994 [39,40]. According to Guilford, 1954, the minimum sample size of 200 is adequate for factor analysis [40]. However, Mundfrom et al., 2005, argue that the number of variables, the number of factors, the number of variables per factor, and the size of the communalities are required for estimating samples sizes for factor analysis [39]. In our case, we used 25 variables (nutrients) to extract four factors (nutrient patterns) and since we used a Principal Component Analysis (PCA) approach of factor analysis, we, therefore, assumed low levels of communality between these variables. According to the sample size calculation tables derived by Mundfrom et al., 2005, a minimum of 160 participants were required [39]. Using the recommendations of both Mundfrom et al., 2005 and Guilford, 1954, a minimum sample size of 200 will have adequate power for principal component analysis [39,40]. Therefore, our sample size of 760 ensured we had sufficient power for this analysis.

## 3. Results

### 3.1. Descriptive Characteristics of the Participants

The participant’s characteristics are presented in Table 1. The mean age in men and women was similar (54 ± 6 years). 61% of participants were currently employed. More men than women were married (*p* = 0.033). Women had a higher mean BMI than men, and a higher proportion of women were classified with obesity (66.5% vs. 21.5%). Accordingly, women had a higher sub-total body fat (%) but had less android (% FM) and more gynoid (% FM) fat than men. While the difference in gynoid percentage fat mass appears small (17.7 (95% CI: 17.4–18.0) vs. 17.0 (95% CI: 16.9–17.2) %FM), this represents a nearly twofold absolute difference in gynoid fat mass between women and men (6.2 vs. 3.2 kg).

VAT and SAT areas were higher in women compared to men, but VAT/SAT ratio was higher in men compared to women.

In terms of dietary intake, men consumed more total energy and a higher proportion of protein and fibre than women. In contrast, women consumed a higher proportion of carbohydrates and fat than men. More men were plausible reporters of dietary energy intake than women. As defined by the mean number of steps per day, physical activity was higher in men than women, but men also reported more sedentary time, expressed as mean sitting time per day. A higher percentage of men were smokers (44.8% and 6.1%, respectively). HIV status and ARV use did not differ by sex.

### 3.2. Nutrient Patterns

The nutrient patterns of men and women were not different (data not shown). Table 2 represents the four nutrient patterns retained after PCA, which cumulatively explained 66.9% of the total variance of the 25 nutrients in the combined sample of men and women. The first PC had high factor loadings for plant protein, starch, B-vitamins, iron and zinc and was termed “Plant Driven Nutrient pattern”. This pattern accounted for 30.2% of the variance in nutrient intake. The second pattern consisted of the high factor loadings of animal protein, cholesterol, fat, fat-soluble vitamins, and vitamin B12. This pattern was named “Animal and Fat Driven Nutrient pattern” and accounted for 17.2% of the variance in nutrient intake. The third PC accounted for 11.2% of the variance in nutrient intake, with the greatest factor loading for vitamin C, sugar, potassium, calcium, and dietary fibre. This pattern was termed “Vitamin C, sugar, Potassium and Calcium Driven Nutrients pattern”. The fourth PC explained 8.1% of the variance in nutrient intake. This PC contained the highest factor loading for retinol and vitamin B12 and was termed “Retinol and Vitamin B12 Driven Nutrient Pattern”.

### 3.3. Associations between Derived Nutrient Patterns with the Selected Body Composition Traits in Men and Women

Associations between the derived nutrient patterns and the selected body composition traits in men and women are presented in Table 3. The plant driven nutrient pattern was not associated with any of the total or regional adiposity measures. However, there was a significant sex*plant driven nutrient pattern interaction for abdominal SAT (P*_int_* = 1.54 × 10^−4^), with the association being significant in women (44.0 cm^2^, 95% CI (21.9; 66.5 cm^2^); *p* = 1.21 × 10^−4^) but not men (–3.0 cm^2^, 95% CI (–9.52; 3.30 cm^2^); *p* = 0.341) as shown in Figure 2.

In the combined sample, we found that the animal and fat driven nutrient pattern was positively and significantly associated with BMI, body fat % and android % FM (Table 3). For BMI only there was a significant sex*animal-driven nutrient pattern interaction (P*_int_* = 0.017), such that the strength of the association was greater in men (1.1 kg/m^2^, 95% CI (0.6–1.7 kg/m^2^), *p* = 1.75 × 10^−5^) compared to women (0.8 kg/m^2^, 95% CI (0.3–1.4 kg/m^2^); *p* = 0.004) (Figure 3).

The Vitamin C, sugar and potassium driven nutrient pattern was positively associated with BMI and body fat% in the combined sample (Table 3). The retinol and vitamin B12 pattern was not associated with total adiposity (BMI or body fat) but was associated with body fat distribution, being positively associated with android FM and VAT, with a tendency to be negatively associated with gynoid fat mass (*p* = 0.054).

## 4. Discussion

Our study set to evaluate the association between nutrient patterns and DXA-derived body fat and regional adiposity in middle-aged black South African (SA) men and women. Four nutrient patterns that explained 67% of the variation in nutrient intake in middle-aged black South African men and women were identified. The plant-driven nutrient pattern explained most of the variance (30%). Although not associated with any of the body fat measures for the combined sample, it was associated with higher abdominal SAT in the women but not the men. The animal and fat driven pattern was associated with greater total and android adiposity. It showed sexual dimorphism in its association with BMI, with the strength of the association being greater in men than women. The vitamin C, sugar and potassium driven nutrient pattern was associated with greater total adiposity as measured by BMI and whole body % fat. In contrast, the retinol and vitamin B12 driven pattern was associated with the centralisation of body fat in the combined sample, with the association with android fat mass being driven by the association in men.

Four studies to date have characterised nutrient patterns of black South Africans. Two of these studies focused on children and adolescents [28,41], while the others focused on adults [29,37]. Similar to our findings, the plant-driven nutrient pattern is the most commonly consumed in South Africa. It is characterised by high factor loadings of magnesium, phosphorus, plant protein, carbohydrates, iron, B-vitamins and fibre [41]. The food sources for this nutrient pattern includes staple foods, such as refined maize meal, which is widely consumed and fortified with B vitamins in South Africa [42]. Although the plant driven nutrient pattern was not significantly associated with total or regional fatness in the combined sample of men and women, we show for the first time that this nutrient pattern was associated with higher abdominal SAT in women but not men. The intake of refined carbohydrates has been associated with increases in abdominal SAT in both men and women [43], but the reasons for the sexual dimorphism in this relationship are not clear. We may postulate that hyperinsulinemia observed in black African women compared to men may drive this relationship [44,45]. Together with the high consumption of processed carbohydrate-rich foods, hyperinsulinemia will drive lipogenesis and SAT accumulation [46]. This may explain this sex-specific relationship and provide a plausible reason why black African women accumulate more abdominal SAT than their male and white European counterparts [44,47]. However, prospective intervention studies are required to verify these suggestions.

The animal protein and fat-driven nutrient pattern, characterised by animal protein, saturated fat, monounsaturated fat, polyunsaturated fat, and cholesterol, was associated with total and central adiposity. This was the second most consumed nutrient pattern and suggested a shift towards a Westernised diet characterised by increased consumption of energy-dense foods, meat, butter, eggs and oils [42], which have been associated with obesity. Notably, we also showed sexual dimorphism in the relationship between animal protein and fat driven nutrient pattern and BMI, showing a stronger relationship in men than women. Socio-cultural perspectives may likely cause this relationship. Traditionally, black SA households are strongly patriarchal, with the nutritional needs of men being prioritised over that of women, particularly relating to protein-rich foods, especially meat [48]. Conversely, one study reported that the young black SA women are also health-conscious and try to minimise foods high in fat [43].

The “vitamin C, sugar and potassium” driven nutrient pattern, which explained 11.7% of the variance in nutrient intake in our sample, suggests a diet high in the consumption of sugar sugar-sweetened beverages, such as carbonated soft drinks fruit juices and tea with milk. The association of added sugars with increased BMI was reported in black South Africans in a longitudinal five-year study and a representative cross-sectional in South African rural and urban areas [49,50]. Similarly, we showed that this nutrient pattern was associated with higher total body fatness in the combined sample and higher abdominal SAT in women. These sex differences might have been due to tea beverages that are more commonly consumed by women than men in this population [42]. Liquid foods, such as sugar-sweetened beverages (SSBs) are less satiating. They have a lower thermic effect than solid foods of similar nutrient composition and energy, leading to higher energy intake, positive energy balance and excess fat accumulation [51].

The retinol and vitamin B12 driven nutrient pattern, suggestive of fish- and animal-based food sources were associated with greater centralisation of body fat (android % FM and VAT) only. Contrary to our results, previous studies reported that vitamin B12 was negatively associated with obesity in Copenhagen adults aged 18 to 60 years [52,53]. However, this was a single nutrient association with obesity, and the association for the combination of retinol and vitamin B12 has not been explored. More research is required to understand this association, which, when stratified by sex, was weakened and only showed a significant association with android % FM in men only.

Our study was limited in that it was a cross-sectional design from which we cannot infer causality. Thus, our study findings will need to be confirmed in prospective randomised clinical trials. Although using self-reported dietary data is a practical tool in all nutritional epidemiological studies, there are limitations around measurement error and final estimation of the nutrient intake. However, the current study used a validated QFFQ, with a list comprising 214 foods eaten by the South African population, which is known to be comprehensive, and was also corrected for energy intake reporting [33]. The strengths of our study included the use of nutrient patterns as opposed to dietary patterns since nutrients are universal and comparable among different ethnicities [28,37]. We used DXA to characterise body fat and regional adiposity objectively. We were interested in the individual associations between nutrient patterns and total and regional adiposity and did not compare the strength of the associations between the body composition measures. Nonetheless, we used Bonferroni correction to account for performing six separate multivariable models. All models were adjusted for energy intake reporting bias that comprised energy intake expenditure derived from objective measures of physical activity (accelerometry). In addition, we adjusted for physical activity and sedentary behaviour (measured objectively using accelerometry), which are known to influence dietary intake and adiposity.

## 5. Conclusions

In conclusion, we showed that nutrient patterns that characterize high intakes of animal protein and fat, as well as vitamin C, sugar, and potassium, were associated with total adiposity, while the animal protein and fat, as well as the B12 driven nutrient patterns, were associated with central adiposity in middle-aged black SA men and women. The study’s novel findings were the sexual dimorphism in the association between the plant-driven nutrient pattern and abdominal SAT and animal and fat driven nutrient pattern with BMI. These findings suggest that Westernization of diets, with increased animal fat and sugar consumption, are associated with higher adiposity and risk for noncommunicable diseases in middle-aged black South Africans. Sexual dimorphism in these associations suggests that men may be more susceptible to obesity when consuming a diet rich in animal protein and fat. In contrast, women may be more susceptible to abdominal SAT accumulation when consuming a staple diet high in processed carbohydrates. Future prospective studies are required to explore the mechanisms underlying these associations, which may provide greater insights into the pathogenesis of obesity and inform sex-specific dietary approaches to curb this increasing burden of disease.

## Figures and Tables

**Figure 1 nutrients-13-04558-f001:**
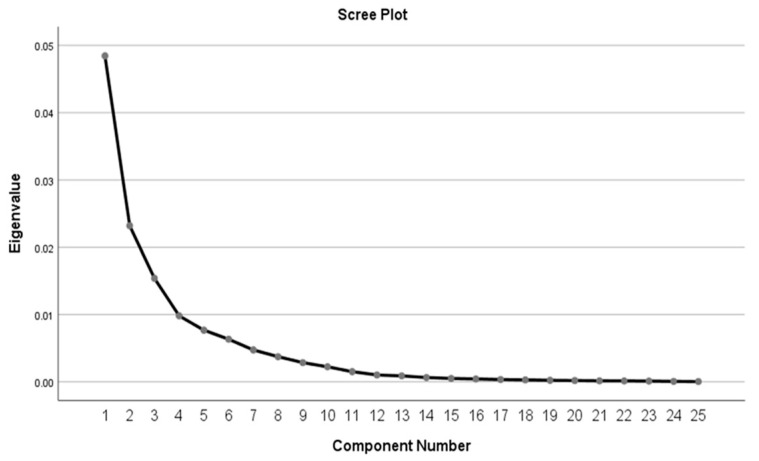
Scree plot showing the eigen values of the principal components which are representative of the nutrient patterns.

**Figure 2 nutrients-13-04558-f002:**
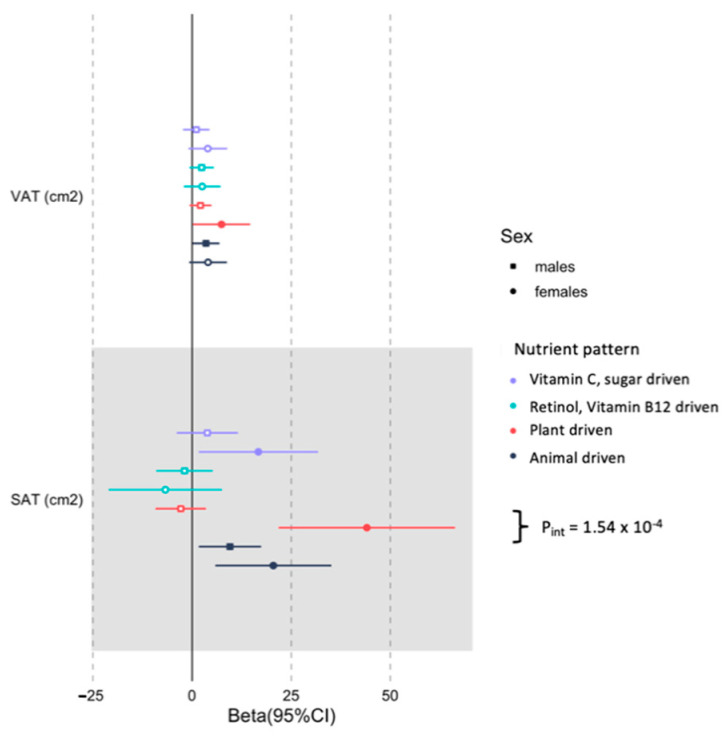
Stratified abdominal subcutaneous adipose tissue (SAT) and visceral adipose tissue (VAT) associations with nutrient patterns in middle-aged black SA men and women. The associations are adjusted for age, number of steps, education status, SES status, energy intake reporting and fat mass index. Non-significant effect sizes are indicated as hollow.

**Figure 3 nutrients-13-04558-f003:**
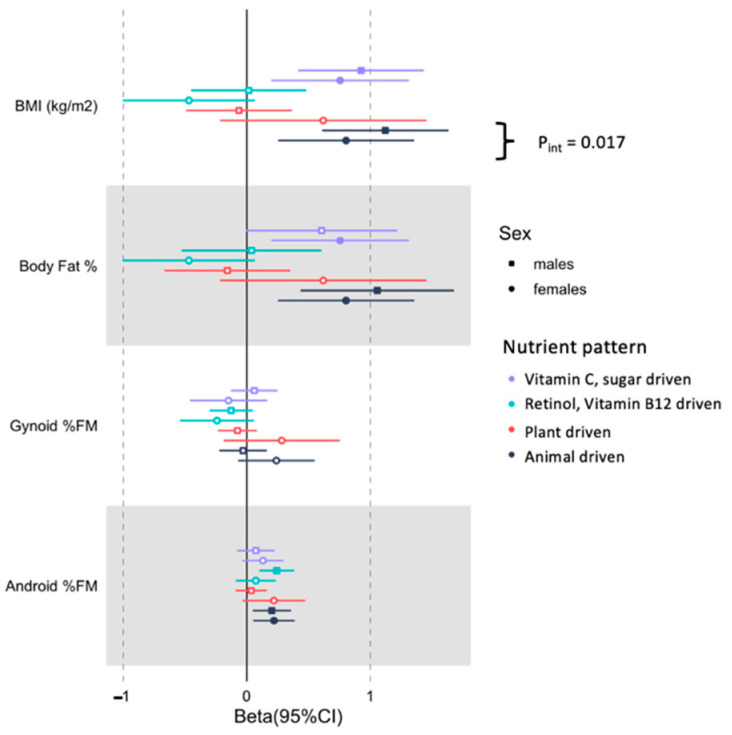
Stratified associations of total and regional adiposity with nutrient patterns in middle-aged black SA men and women. The associations are adjusted for age, number of steps, education status, SES status and energy intake reporting. A significant interaction between sex and animal-driven nutrient pattern on BMI was noted (P*_int_* = 0.017). Non-significant effect sizes are indicated as hollow.

**Table 1 nutrients-13-04558-t001:** Participant characteristics.

Variables	Men (*n* = 414)	Women (*n* = 346)	*p*-Value
Age (yrs)	54 ± 6	54 ± 6	0.817
Measure of Ses			
Education (*n* (%))			
Primary	101 (24.5)	69 (20.1)	<0.001
Secondary	231 (56.1)	239 (68.3)	
Tertiary	80 (19.4)	40 (11.6)	
% Employed (*n* (%))	255 (61.7)	212 (61.4)	0.934
% Married (*n* (%))	208 (50.4)	147 (42.6)	0.033
BMI (kg/m^2^)	25.5 ± 5.9	33.2 ± 6.5	<0.001
BMI categories (*n* (%))			
Underweight	38 (9.2)	2 (0.6)	<0.001
Normal weight	167 (40.4)	32 (9.2)	
Overweight	119 (28.8)	82 (23.7)	
Obese	89 (21.5)	230 (66.5)	
Total and Regional Adiposity			
Fat mass (kg)	18.9 ± 8.9	35.5 ± 10.2	<0.001
Body fat (%)	26.0 ± 6.8	44.6 ± 5.2	<0.001
Gynoid (% FM)	17.0 ± 1.9	17.7 ± 2.7	<0.001
Android (% FM)	8.5 ± 1.6	7.4 ± 1.5	<0.001
VAT (cm^2^)	87.4 ± 46.0	104.1 ± 44.3	<0.001
SAT (cm^2^)	311 ± 192	460 ± 155	<0.001
VAT/SAT ratio	1 ± 0	0 ± 0	<0.001
Dietary Intake			
Energy intake (kj)	8691 ± 4192	6960 ± 2923	<0.001
Carbohydrates (% EI)	53.8 ± 9.3	56.3 ± 8.4	<0.001
Protein (% EI)	12.1 ± 3.0	11.5 ± 2.6	0.009
Fat (% EI)	28.9 ± 7.2	30.9 ± 7.0	<0.001
Fibre (g)	19.9 ± 9.4	17.6 ± 8.5	0.001
Lifestyle Factors			
Number of steps (×1000)	10.6 ± 4.7	9.2 ± 3.7	<0.001
Sitting time (hours)	7.8 ± 1.9	7.1 ± 1.9	<0.001
% Smokers (*n* (%))	185 (44.8)	21 (6.1)	<0.001
% HIV Positive (*n* (%))	86 (20.9)	66 (19.1)	0.527
% ARVs	75 (92.8)	53 (93.0)	0.931
Dietary energy reporting (*n* (%))			
Underreporting	176 (42.5)	244 (70.5)	<0.001
Over reporting	34 (8.2)	7 (2.0)	
Plausible reporters	204 (49.3)	95 (27.5)	

Values are presented as mean ± standard deviation or count (percentage). BMI, body mass index; FM, sub-total fat mass; VAT, visceral adipose tissue; SAT, subcutaneous adipose tissue; %EI, percentage of total energy intake; HIV, human immunodeficiency virus; ARV, antiretroviral therapy.

**Table 2 nutrients-13-04558-t002:** Nutrient patterns and factor loadings for the combined sample of men and women.

Nutrients	Plant Driven Nutrient Pattern	Animal Protein and Fat Driven Nutrient Pattern	Vitamin C, Sugar and Potassium Driven Nutrient Pattern	Retinol and Vitamin B12 Driven Nutrient Pattern
Plant protein	**0.821**	0.116	0.122	−0.056
Animal protein	0.131	**0.725**	0.175	0.243
Saturated fat	0.315	**0.661**	0.206	0.077
Monounsaturated fat	0.296	**0.712**	0.156	−0.017
Polyunsaturated fat	**0.613**	**0.565**	0.019	−0.064
Cholesterol	0.095	**0.769**	−0.020	0.463
Starch	**0.799**	0.092	−0.167	−0.042
Sugar	0.021	−0.046	**0.726**	0.033
Dietary Fibre	**0.632**	0.063	**0.477**	−0.047
Calcium	0.220	0.224	**0.555**	0.287
Iron	**0.856**	0.295	0.241	0.120
Magnesium	**0.795**	0.135	0.259	0.056
Phosphorus	**0.739**	0.301	0.147	0.142
Potassium	0.318	0.079	**0.653**	0.075
Zinc	**0.852**	0.350	0.173	0.073
Retinol	0.080	0.206	0.130	**0.960**
Beta carotene	0.008	0.058	0.279	−0.017
Thiamine	**0.901**	0.287	0.221	0.012
Riboflavin	**0.754**	0.408	0.252	0.307
Vitamin B6	**0.674**	0.082	0.033	0.014
Folate	**0.748**	0.025	0.067	0.402
Vitamin B12	0.069	**0.496**	0.073	**0.636**
Vitamin C	0.094	0.181	**0.888**	−0.019
Vitamin D	0.064	**0.753**	−0.048	0.205
Vitamin E	0.256	**0.610**	0.030	−0.036
Explained variance %	30.287	17.202	11.263	8.199
Cumulative explained variance %	30.287	47.490	58.753	66.952

Bold factor loadings used to indicate factor loadings >±0.47 for naming nutrient patterns.

**Table 3 nutrients-13-04558-t003:** Regression coefficients for 1 SD increase in the derived nutrient pattern scores for selected body composition traits.

	BMI		Body Fat %		Gynoid Fat %		Android Fat %		VAT (cm^2^)		SAT(cm^2^)	
	B (95% CI)	*p*	B (95% CI)	*p*	B (95% CI)	*p*	B (95% CI)	*p*	B (95% CI)	*p*	B (95% CI)	*p*
Plant Driven Nutrient pattern	0.39 (−0.02; 0.80)	0.065	0.05(−0.37; 0.46)	0.831	−0.02(−0.1; 0.149)	0.785	0.08(−0.03; 0.19)	0.153	1.12 (−1.28; 3.53)	0.360	−1.10(−5.18; 2.99)	0.598
Animal protein and Fat Driven Nutrient pattern	0.80 (0.40; 1.20)	<0.001	0.91(0.50; 1.32)	<0.001	0.08(−0.10; 0.25)	0.382	0.21(0.10; 0.32)	<0.001	1.519 (−0.90; 3.94)	0.218	2.37(−1.73; 6.47)	0.257
Vitamin C, sugar and potassium Driven Nutrient pattern	0.99 (0.59; 1.39)	<0.001	0.74(0.32; 1.15)	<0.001	−0.02(−0.19; 0.16)	0.866	0.99 (−0.01; 0.21)	0.081	0.79(−1.63; 3.21)	0.522	0.83(−3.27; 4.93)	0.692
Retinol and Vitamin B12 Driven Nutrient pattern	0.44 (−0.34; 0.43)	0.819	−0.09 (−0.48; 0.31)	0.672	−0.16(−0.32; 0.003)	0.054	0.19(0.08; 0.30)	<0.001	4.15 (1.86; 6.44)	<0.001	3.82(−0.07; 7.70)	0.054
Dietary energy intake reporting												
Underreporting	5.65 (4.75; 6.54)	<0.001	4.35(3.44; 5.27)	<0.001	−0.87(−1.25; −0.50)	<0.001	0.76(0.51; 1.00)	<0.001	−4.97 (−10.81; 7.46)	0.095	1.62(−17.99; 21.22)	0.872
Over reporting	−2.89 (−4.67; −1.12)	<0.001	−3.46(−5.28;−1.65)	<0.001	0.07(−0.68; 0.82)	0.855	−0.36(−0.85; 0.13)	0.154	−8.93 (−20.49; 2.64)	0.130	3.83(−6.08; 13.73)	0.449
Plausible reporting (reference)												
Age	0.02 (−0.05; 0.08)	0.648	0.10(0.03; 0.17)	0.005	−0.01(−0.04; 0.01)	0.318	0.03(0.01; 0.04)	0.006	0.92 (0.52; 1.31)	6.85 × 10^−6^	0.12(−0.55; 0.79)	0.731
Sex (Male; female reference)	−5.80 (−6.65; −4.94)	<0.001	−16.97(−17.85;−16.09)	<0.001	−1.07(−1.43;−0.70)	<0.001	1.35(1.12; 1.16)	<0.001	−39.02(−45.47; −32.58)	<0.001	−35.19(−46.12; −24.26)	<0.001
Number of steps (×1000)	−0.25 (−0.35; −0.16)	<0.001	−0.27(-0.37; −0.17)	<0.001	0.08(0.04; 0.12)	<0.001	−0.06(−0.08; −0.03)	<0.001	−0.70 (−1.30; −0.10)	0.021	−0.05(−1.07; 0.96)	0.917
Sitting time (h)	0.28(0.07; 0.50)	0.011	0.12(−0.11; 0.34)	0.303	0.05(−0.04; 0.15)	0.274	−0.002(−0.06; 0.06)	0.955	−1.13 (−2.44; −0.19)	0.092	−1.89(−4.11; 0.34)	0.098
Education												
Primary	0.88 (−0.41; 2.18)	0.181	0.04(−1.30; 1.38)	0.952	−0.29(−0.85; 0.26)	0.301	0.002(−0.36; 0.36)	0.989	1.80 (−3.97; 7.58)	0.541	−7.18(−17.54; 9.05)	0.151
Secondary	−0.02 (−1.11; 1.06)	0.967	−0.07(−1.19; 1.06)	0.904	0.01(−0.45; 0.48)	0.962	0.06(−0.24; 0.37)	0.677	−0.38 (−8.22; 7.46)	0.924	−4.25(−16.97; 2.72)	0.531
Tertiary (reference)												
Body fat (kg)									3.18 (2.89; 3.46)	<0.001	13.24(12.76; 13.72)	<0.001
Unadjusted R2	0.478		0.771		0.089			0.212		0.532		0.871
Adjusted R2	0.469		0.767		0.074			0.199		0.501		0.863

SD, standard deviation; CI, confidence interval; EI, energy intake; EE, energy expenditure.

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
