# Peer review of "Sex Differences in the Associations of Nutrient Patterns with Total and Regional Adiposity: A Study of Middle-Aged Black South African Men and Women"

_nutrients, 2021, doi:10.3390/nu13124558_

Round 1

Reviewer 1 Report

In this manuscript, Ratshikombo et al., demonstrated how nutrient patterns influenced total and regional adiposity in men and women populations. It was a clear study with nice technical details.

Author Response

Thank you for the positive feedback.

Reviewer 2 Report

This article tried to differentiate the sex difference on adiposity under the four food patterns in middle-aged black South African and concluded that some food patterns have a different influence between males and females. However, the retrospective study may not demonstrate a clinically meaningful result with the reasons listed below

  1. The main method referenced/ used (Hu,2002) was not “peer-reviewed” (preprint)
  2. Results obtained by a single method should be validated or supported by “traditional approaches”
  3. The animal-driven nutrient or the plant-driven pattern induced gender differences that seemed irrelevant to general clinical observation.  
  4. Data analysis showed “vague” results (example: Table 1. Gynoid (%FM 17.0±1.9 vs 17.7±2.7, resulted in a p-value of <0.001), making all data sets unreliable.

Suggestion: authors should design additional prospective study(s) to support or validate the conclusions.

The article shows some errors in the text, only exampling to the Abstract section.

  1. Line 13 patterns and body fat or regional…
  2. Line 14 the middle-aged black…
  3. Line 19 The animal and fat….
  4. Line 22 than in women…
  5. Line 24 44 kg/cm2; but not in men…
  6. Line 25 “interactions” should be replaced, suggesting to” differences”; missing (??kg/cm2,95% CI)
  7. Line 26 the associations should be their associations

Author Response

Thank you for the feedback, below are the reviewer’s comments and the authors response

  1. The main method referenced used (Hu, 2002) was not a “peer-reviewed” (preprint)

My apologies for the systematic technical errors due to my reference manager which added preprint to my references. This has been corrected. (See reference list). The main method reference is a published paper from 2002, and has been cited 1,155 times to date, highlighting its relevance.

  1. Results obtained by a single method should be validated or supported by “traditional approaches “

As mentioned above, the methodology used to capture nutrient patterns has been extensively used both globally and within the South African context, examples include. PMID: 32393204; PMID: 27764841; PMID: 28106816. We therefore feel there is no need for to use multiple approaches to validate this method.

  1. The animal-driven nutrient or plat driven pattern induced gender differences that seemed irrelevant to general clinical observation.

Although we acknowledge the reviewers’ point that the clinical significance of the sex differences in the relationships between the nutrient pattens are not obvious in this cross-sectional analysis, the study provides critical insights into sex disparities in risk, which can be validated in longitudinal or intervention studies. These findings will provide guidance that is clinically relevant for the dietary management of obesity in black SA men and women.

  1. Data analysis showed “vague” results (example: Table 1. Gynoid (%FM 17.0±1.9 vs 17.7±2.7, resulting in a p-value of < 0.001), making all data sets unreliable.

We acknowledge the reviewer’s concern, but although it may appear that this is small difference in gynoid percentage fat mass between men and women, this difference was very consistent and considering the vast differences in total fat mass (with nearly a twofold absolute difference in gynoid fat mass between women and men, 6.2 vs. 3.2 kg), we believe the results are valid and not “vague” as the reviewer suggests.

Suggestion: authors should design additional prospective study(s) to support or validate the conclusion.

-Thank you for this comment it’s an excellent addition, we have added it in the conclusion.

The article shows some errors in the text, only exampling to the Abstract section.

  1. Line 13 patterns and body fat of regional…

Thank you, we have corrected the tense.

  1. Line 14 the middle-aged black…

We used middle-aged consistently throughout the paper

  1. Line 19 The animal and fat…

This has been change to ensure consistency to animal driven

  1. Line 22 than in women…

We have added “in” as suggested.

  1. Line 24 44kg/cm2; but not in men

Sorry for the typo, SAT is measured in cm2 and has been corrected throughout.

  1. Line 25 “interections” should be replaced, suggesting to “differences”; (??kg’cm2, 95%CI)

Thank you, we have corrected this.

  1. Line 26 the association should be their association

Thank you, this has been corrected.

Reviewer 3 Report

This is a well-written manuscript. Below are my comments

Introduction

This is a well written introduction. This is a minor suggestion. 

Line 77-78- just write that you are unaware of any studies instead of a definitive that there are no studies

Methods

You did a good job with the methodology. Below are my comments to help make it easier to replicate.

  1. How was the sample size of the sub-sample determined?
  2. I couldn't find Figure 1 in the manuscript (maybe there was an editorial error)
  3. Besides employment status how was socioeconomic status determined?
  4. Since this is not reported in the results section, you can report this in the methodology. Please report mean+/- SD for number of days that the accelerometer was worn for. Same with mean+/- SD for the number of hours the accelerometer was worn.
  5. I know that ActivePAL has been validated as a reliable measure of PA. Can you cite that it has been validated
  6. Was all data normally distributed? If not, were any techniques used to normalize the data? With this large sample you don't have to normalize the data, but please make sure you mention whether any data was not normally distributed and if it was not, what normalization techniques were applied.
  7. Why did the researchers choose steps as a co-variate instead of intensity of PA? I know that ActivePAL has the ability to provide you with that information
  8. Did you include SES in your analysis?
  9. Considering you were examining 4 dependent variables using the same independent variables, why did the researchers not utilize a multi-variate multiple regression? This would allow you to determine the difference in strength of relationship for each of the independent variables between the 4 dependent variables (i.e. Vitamin C intake is more important for BMI than body fat even though both are significant/there is no difference in importance for Vitamin C between BMI and body fat). It would also cut down on multiple analyses and the need to perform a correction for multiple analyses. As your analysis sits right now you may have a false positive. Therefore, I would highly recommend the authors use a multi-variate multiple regression as it allows you to simultaneously test for multiple associations between the four dependent variables. 
  10. Figure 2 needs to be clearly labeled (might be an editorial issue)

Results

The results are very well-written. However, the authors should use a multi-variate multiple regression. Until then I will comment on what is currently in the manuscript.

  1. Table 2 was very informative
  2. Figures 3 and 4 were also very informative
  3. I think that by conducting the proper analyses you will be able to provide more insightful results.

Discussion

I would start the discussion by stating how your results fulfilled the aims of your study rather than starting with how the 25 nutrients were broken down into 4 patterns. 

Other than that I cannot properly evaluate the discussion until the statistical analysis section is updated. 

Author Response

Thank you for the feedback, below are the reviewer’s comments and the authors response

  1. How was the sample size of the sub-sample determined?

Sample size estimation of factor analysis is a highly debated topic (Gaskin and Happell, 2014). However, we estimated the sample size using the guidelines recommended by Kline, 1994 and Mundfrom et al. 2005. According to Kline, 1994 the minimum sample size of 200 is adequate for factor analysis.  However, Mundfrom et al. 2005, argues that the number of variables, the number of factors, the number of variables per factor, and the size of the communalities are required for estimating samples sizes for factor analysis. In our case we used 25 variables (nutrients) to extract four factors (nutrient patterns) and since we used a Principal Component Analysis (PCA) approach of factor analysis, we therefore assumed low levels of communality between these variables. According to the sample size calculation tables derived by Mundfrom et al. 2005, a minimum of 160 participants were required. Using the recommendations of both Mundfrom et al 2005 and Kline, 1994, we chose a sample size of 200. Therefore, our sample size of 760 ensured we had sufficient power for this analysis.

See line 206 to 219

  1. I couldn’t find figure 1 in the manuscript (maybe there was an editorial error).

Apologies there is no figure 1, this has been corrected.See line 89

  1. Besides employment status how was socioeconomic status determined?

We used education as a proxy for socioeconomic status as it was significantly differently in men and women, and therefore used as a covariate in the regression models. In contrast, employment status, which was our other measure of socioeconomic status, was not different between men and women and therefore not included in the regression models.  (see table 1).

  1. Since this is not reported in the results section, you can report this in the methodology. Please report mean± SD for number of hours the accelerometer was won.

We have reported that the ActivPAL was won one average for five to seven days (5.93 ±0.859). See line 120

  1. I know that ActivePal has been validated as a reliable measure of PA. can you cite that it has been validated

The citation was added see line 118.

  1. Was all data normally distributed? If not, were any techniques used to normalize the data? With this large sample you don’t have to normalize the data, but please make sure you mention whether any data was normally distributed and if it was not, what normalization techniques were applied.

We did normality tests as indicated in line 158 and not all the data was normally distributed. For  descriptive data that was skewed, we used non parametric tests,  and parametric test were used for the data that was normally distributed ( See Table 1). However, for the linear models we agree with the reviewer, and we did not transform the data due to the central limit theorem.

  1. Why did the researchers choose steps as a co-variate instead of intensity of PA? I know that ActivePal has the ability to provide you with that information

As ActivePal is a better measure of steps and sedentary behaviours than PA intensity (Wu et al 2021), we chose to use the steps and sitting time as covariates in the models. Further, most of the activity was done at a low intensity, typically as walking for travelling, with very limited lifestyle leisure activity.

Reference

Wu Y, Johns JA, Poitras J, Kimmerly DS, O'Brien MW. Improving the criterion validity of the activPAL in determining physical activity intensity during laboratory and free-living conditions. J Sports Sci. 2021 Apr;39(7):826-834. doi: 10.1080/02640414.2020.1847503

  1. Did you include SES in you analysis?

Yes, we included education as a proxy for SES as it was significantly different between men and women, as illustrated in Table 1.

  1. Considering you were examining 4 dependent variables using the same dependent variables, why did the researchers not utilize a multi-variable regression? This would allow you to determine the difference in strength of relationship for each of the independent variables between the 4 dependent variables _i.e. Vitamin C intake is more important for Vitamin C between BMI and body fat). It would also cut down on multiple analyses and the need to perform a correction for multiple analyses. As your analysis sits right now you may have false positive. Therefore, I would highly recommend the authors use a multi-variate multiple regression as it allows you to simultaneously test for multiple associations between the four dependent variables.

The multivariable models included the body composition variables as dependent (outcome) variables with the nutrient patterns included as independent variables in the same model, with the addition of covariates. Hence a multivariable approach (as suggested by the reviewer) was used. Only sex regression models were run for the body composition outcomes (BMI, body fat %, Gynoid %, Android %, VAT and SAT).  In order to clarify the analysis strategy, we have made changes to the description of the method on lines 196-200, which now reads as follows: “The nutrient patterns were included as predictors in the same model, and the covariates included sex, age, number of steps, sitting time, dietary energy intake reporting status (under, plausible and over-reporters), educational achievement, marital status, smoking, and HIV status

  1. Figure 2 needs to be clearly labeled (might be an editorial issue)

We have added a legend to Figure 2 . See line 190  

Results

The results are very well-written. However, the authors should use a multi-variate multiple regression. Until then I will comment on what is currently in the manuscript.

  1. Table 2 was very informative

Thank you

  1. Figures 3 and 4 were also very informative

Thank you

  1. I think that by conducting the proper analyses you will be able to provide more insightful results.

Discussion

I would start the discussion by stating how your results fulfilled the aims of your study rather than starting with how the 25 nutrients were broken down into 4 patterns. 

We have revised the opening of the discussion to start with the aim of the study. See line 314-317

Round 2

Reviewer 2 Report

Data validation is required.

Author Response

Data validation is required.

As mentioned previously, the methodology used to capture nutrient patterns has been extensively used both globally and within the South African context, examples its not novel so as to necessitate  additional validation .  333 papers have been published to date focusing on the nutrient patterns  which  we used in our study,according to PubMed (https://www.ncbi.nlm.nih.gov/pmc/?term=%22Nutrient+patterns%22 ). Some notable related papers include. PMID: 32393204; PMID: 27764841; PMID: 28106816.We therefore feel there is no need for to use multiple approaches to validate this method.

Reviewer 3 Report

Thank you for addressing many of my comments.

The one comment that you did not address was that you did not perform a multi-variate multiple regression. A multi-variate multiple regression allows you to examine the interaction between multiple independent and dependent variables. Considering you have 6 dependent variables and you'd like to examine the associations between each of them and 12 independent variables, a multi-variate multiple regression model would be the ideal way to analyze this data. Additionally, this analysis would allow you to determine the strength of differences in the association value between the independent variable the the multiple dependent variables that it was associated with. For example, animal protein is associated with BMI, Body Fat and Android Fat. A multivariate multiple regression would allow you to determine whether the strength of association for animal fat with BMI was > or < than that with Body Fat. In these sorts of analyses it is difficult to use beta values due to the differences in the CI's to determine whether the strength of association was > or <. 

Additionally, were all the models significant? Please provide R^2 and adjusted R^2 values. Also with 6 different regression models, did you correct for multiple analyses?

Author Response

On behalf of all the co-authors, I would like to thank the reviewers for their constructive comments on the manuscript.  We have studied them in detail and have responded to each of them below, with the changes to the manuscript indicated as tracked changes.  We are of the opinion that these comments have significantly improved the paper and hope that it is now suitable for publication.

  1. One comment that you did not address was that you did not perform a multi-variate multiple regression. A multi-variate multiple regression allows you to examine the interaction between multiple independent and dependent variables. Considering you have 6 dependent variables and you'd like to examine the associations between each of them and 12 independent variables, a multi-variate multiple regression model would be the ideal way to analyze this data. Additionally, this analysis would allow you to determine the strength of differences in the association value between the independent variable the the multiple dependent variables that it was associated with. For example, animal protein is associated with BMI, Body Fat and Android Fat. A multivariate multiple regression would allow you to determine whether the strength of association for animal fat with BMI was > or < than that with Body Fat. In these sorts of analyses it is difficult to use beta values due to the differences in the CI's to determine whether the strength of association was > or <. 

A multivariate multiple regression  seeks to model the linear relationship between more than one independent variable (IV) and more than one dependent variable (DV). In our work we are mainly interested in the sex differences of the associations of the nutrient patterns with the body composition outcomes separately. That’s why we tested these outcomes separately to see which of these had significant sex interactions with the nutrient patterns. The reviewers suggestions are noted . However,  our aim of the study was not to assess which outcome associated strongly with a particular nutrient pattern .But whether there are sex differences in the associations of these nutrient patterns with the individual outcomes.

  1. Additionally, were all the models significant? Please provide R^2 and adjusted R^2 values.

All the models were significant (p < 0.001). The R^2 and adjusted R^2 values have been added to Table 3.

  1. Also with 6 different regression models, did you correct for multiple analyses?

We note the reviewers comment about the need for multiple testing. But we think this relates to the comment above for the multi-variate multiple regression. In our study we are considering this outcomes as independent from each other. Since we are interested to evaluate sex differences for the association of each outcome variable independent of the other therefore we feel the multiple testing correction might not be necessary. Since our aim is not to model the linear relationship between more than one independent variable (IV) and more than one dependent variable (DV).  

Nonetheless,  using the Bonferroni correction for multiple testing we consider a significant p-value of 0.05/6 =0.008. All our nutrient pattern associations in Table 3 that all have p values <0.001 are still significant using the multiple testing correction.